# Torsional periodic lattice distortions and diffraction of twisted 2D materials

Suk Hyun Sung [1], Yin Min Goh [2], Hyobin Yoo [3], Rebecca Engelke [4], Hongchao Xie [2], Kuan Zhang[5], Zidong Li [6], Andrew Ye[7], Parag B. Deotare [6,8], Ellad B. Tadmor[5], Andrew J. Mannix [9], Jiwoong Park [7,10], Liuyan Zhao [2], Philip Kim [4] & Robert Hovden [1,8] ✉

Twisted 2D materials form complex moiré structures that spontaneously reduce symmetry through picoscale deformation within a mesoscale lattice. We show twisted 2D materials contain a torsional displacement field comprised of three transverse periodic lattice distortions (PLD). The torsional PLD amplitude provides a single order parameter that concisely describes the structural complexity of twisted bilayer moirés. Moreover, the structure and amplitude of a torsional periodic lattice distortion is quantifiable using rudimentary electron diffraction methods sensitive to reciprocal space. In twisted bilayer graphene, the torsional PLD begins to form at angles below 3.89° and the amplitude reaches 8 pm around the magic angle of 1.1°. At extremely low twist angles (e.g. below 0.25°) the amplitude increases and additional PLD harmonics arise to expand Bernal stacked domains separated by well defined solitonic boundaries. The torsional distortion field in twisted bilayer graphene is analytically described and has an upper bound of 22.6 pm. Similar torsional distortions are observed in twisted $WS_2$, $CrI_3$, and $WSe_2/MoSe_2$.

Periodic lattice distortions (PLD) are at the heart of correlated electronic behavior such as superconductivity[1], metal-insulator transitions[2], and charge density waves (CDW)[3]. PLDs are typically intrinsic to a crystal[3,4], Fermi-surface driven[5], accompanied by a CDW, and have periodicity spanning a few unit cells (~1–2 nm). However, recently extrinsic van der Waals (vdW) driven superlattices with tunable periodicity (up to a few 100 nm) were discovered in twisted bilayer graphene (TBG)[6]. TBG has been spotlighted for extraordinary correlated electron behaviors for a twist at the so-called "magic" angle (~1.1°)[7]. Yoo et al. showed that magic angle TBG is not a simple superposition of two constituent layers[6], but rather a 2D crystal that periodically restructures at the mesoscale. Subsequent reports showed moiré superlattices of other vdW systems with similar periodic restructuring[8–11]. Furthermore, this restructuring has a dramatic effect on the band structure, magnetism, and superconducting properties[6,9,12]. Therefore, understanding twisted 2D materials requires a full description of the atomistic structure down to picoscale displacements. However, a systematic depiction of restructured moiré superlattices is nearly absent and limited to descriptions of local stacking geometry.

Here, we show the atomic structure of 2D moiré superlattices at and near the magic angle are concisely and accurately described by a torsional PLD comprised of three transverse displacement waves. In this way, the complexity of low-twist moiré crystal restructuring is reduced to a single PLD order parameter with an amplitude and wave vector. Each layer in the bilayer system has an equal and opposite

[1]Department of Materials Science and Engineering, University of Michigan, Ann Arbor, MI 48109, USA. [2]Department of Physics, University of Michigan, Ann Arbor, MI 48109, USA. [3]Department of Physics, Sogang University, Seoul 04107, Republic of Korea. [4]Department of Physics, Harvard University, Cambridge, MA 02138, USA. [5]Aerospace Engineering and Mechanics, University of Minnesota, Minneapolis, MN 55455, USA. [6]Electrical and Computer Engineering Department, University of Michigan, Ann Arbor, MI 48109, USA. [7]Pritzker School of Molecular Engineering, University of Chicago, Chicago, IL 60637, USA. [8]Applied Physics Program, University of Michigan, Ann Arbor, MI 48109, USA. [9]Department of Materials Science and Engineering, Stanford University, Stanford, CA 94305, USA. [10]Department of Chemistry, University of Chicago, Chicago, IL 60637, USA. ✉e-mail: hovden@umich.edu

torsional PLD amplitude. From quantitative diffraction of low twist angle and magic angle graphene, the atomic displacements of the larger superlattice can be measured. In twisted bilayer graphene, we report a torsional PLD amplitude of 7.8 ± 0.6 pm and 6.1 ± 0.4 pm for twist angle ($\theta$) of 1.1° and 1.2°, respectively. We report an upper bound for the PLD amplitude of twisted bilayer graphene to be 22.6 pm based on interlayer interaction energy. In addition, we show that the torsional PLD amplitudes can be accurately predicted across all twist angles using an analytic model of the vdW stacking and elastic energies. Lastly, we show that this torsional PLD exists across a variety of other layered 2D materials.

## Results

### Periodic restructuring in twisted bilayer graphene

Moiré patterns emerge for two rotated lattices. In TBG, two single layers of graphene are stacked with a small interlayer twist (Fig. 1c). The moiré of this twisted bilayer graphene (TBG) is an alternating pattern of three high symmetry stackings (AA, AB, and BA), separated by channels of solitonic, intermediate dislocation (often described as an energetic saddle-point)[13,14]. In the energetically favorable AB stacking (also called Bernal stacking), half the atoms in one layer are atop atoms in the layer below (Fig. 1d); BA stacking is the mirror of AB stacking. AA stacking (Fig. 1e), where all atoms in both layers are aligned, requires much higher energy (~19 meV/atom[15]). Despite the complex super-structure of moiré stacking, the diffraction pattern of TBG is a simple superposition of two rotated single-layer Bragg peaks[16]—validated by quantum mechanical scattering simulation in Fig. 2b and previously measured experimentally at higher twist angles[17].

In low twist angle bilayer materials, a striking restructuring of the moiré lattice emerges. Dark-field (DF-) TEM[6,13] and later 4D-STEM[18,19] revealed that this superlattice corresponds to a triangular array of AB/BA domains. However, domain boundaries soften near or above the magic angle ($\theta \approx 1°$) and a simple array of perfect AB/BA domains fails to correctly capture the full atomistic structure of the twisted system[20,21].

Here, we show that a PLD model provides a precise and concise description of lattice restructuring in TBG. PLDs are sinusoidal displacements of atomic positions ($\mathbf{r}' = \mathbf{r}_0 + \mathbf{A}\sin(\mathbf{q} \cdot \mathbf{r}_0)$; $\mathbf{r}'$, $\mathbf{r}_0$ are deformed and original atomic positions, $\mathbf{q}$ is the wave vector, and $\mathbf{A}$ is the displacement vector). Both longitudinal ($\mathbf{A}\|\mathbf{q}$) and transverse ($\mathbf{A}\perp\mathbf{q}$) distortion waves naturally emerge in various charge-ordered

crystals, including 2D materials (e.g., longitudinal: TaS$_2$[3,22]; NbSe$_2$[23]; or transverse: BSCMO[4], UPt$_2$Si$_2$[24]).

A torsional PLD succinctly and accurately describes the relaxed structure of TBG (Fig. 1b). The torsional displacement field is made from three non-orthogonal, transverse PLDs (Fig. 1a):

$$\Delta_n = A_n \sum_{i=1}^{3} \hat{\mathbf{A}}_i \sin(n\mathbf{q}_i \cdot \mathbf{r}_0 + \phi_i); \quad \hat{\mathbf{A}}_i \perp \mathbf{q}_i \qquad (1)$$

Here, $\mathbf{r}_0$ are undistorted atom positions, $\mathbf{q}_i$ is the PLD wave vector, and $\hat{\mathbf{A}}_i$ is the unit vector describing the transversity of the PLD. The distorted lattice positions are given by: $\mathbf{r} = \mathbf{r}_0 + \Delta_n$. Three $\mathbf{q}$'s are 120° apart with a magnitude set by the twist angle ($|\mathbf{q}| \approx b\theta$, $b$ is the reciprocal lattice constant) to accommodate the symmetry of the moiré pattern (See Supplementary Note 1). The phase, $\phi_i$, shifts or alters the relaxation patterns (Supplementary Fig. S8). For TBG, the origin is placed at the AA center ($\phi_i = 0$). Importantly, this torsional wave occurs in both layers, however, the direction of the field in each layer is reversed such that distortions are opposite. Transverse distortions, $\hat{\mathbf{A}}_i \perp \mathbf{q}_i$, are expected when the lattice constants of both layers are equivalent (otherwise longitudinal components, $\hat{\mathbf{A}}_i \| \mathbf{q}_i$, may be present). A single torsional PLD ($\Delta_1$) is typically sufficient to describe the system, however, more generally, PLDs with higher-order harmonics ($n > 1$) are permissible and the total displacements become the sum of multiple harmonics (as discussed later).

Figure 1a illustrates the displacement field ($\Delta_1$) from a torsional PLD in one layer of twisted bilayer graphene. The arrows show the direction and magnitude with which atoms displace from their expected lattice sites. The torsional field is a nanoscale trigonal lattice of rotational distortions spaced $1/|\mathbf{q}|$ apart. The distortion field exhibits behaviors desired for relaxation of TBG—twisting AA regions in one direction and anti-twisting AB/BA regions in the other. The vdW interaction between layers strives to locally twist (anti-twist) AA (AB/BA) regions to minimize (maximize) interlayer registration and reduce the total interaction energy. The relaxed structure (Fig. 1b)—obtained by applying displacements to original atomic sites (Fig. 1c)—acts to maximize the low-energy regions with AB/BA stacking and decreases the high-energy regions with AA stacking.

Torsional PLDs are immediately apparent in an electron diffraction pattern. This atomic restructuring manifests as superlattice peaks

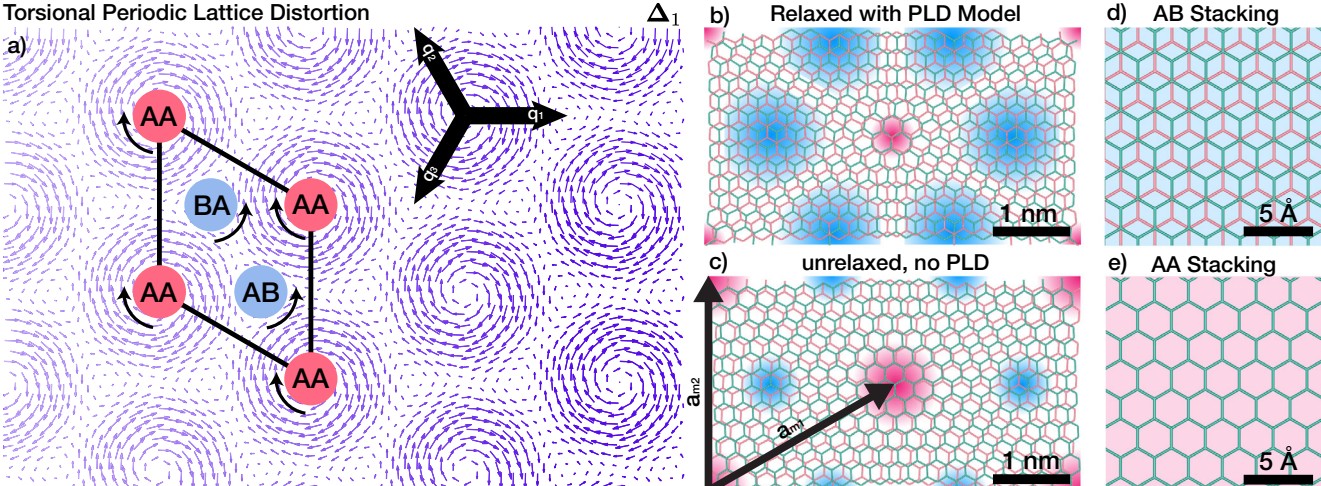

**Fig. 1 | Periodic restructuring of twisted bilayer graphene (TBG). a** Displacement field of torsional PLD. Local rotational fields for the AA region and AB/BA regions are opposite. By including higher harmonics, PLD can exhibit any arbitrary pattern. The moiré supercell crystal structures of **c** pristine TBG and **b** restructured TBG with torsional PLD model. Red and blue overlay highlights energetically unfavorable AA stacked region, and stable AB/BA stacked region, respectively. PLD decreases the total energy of the system by expanding AB/BA domain and decreasing the AA domain. Crystal structure of **d** AB stacked and **e** AA stacked bilayer graphene are shown as a reference.

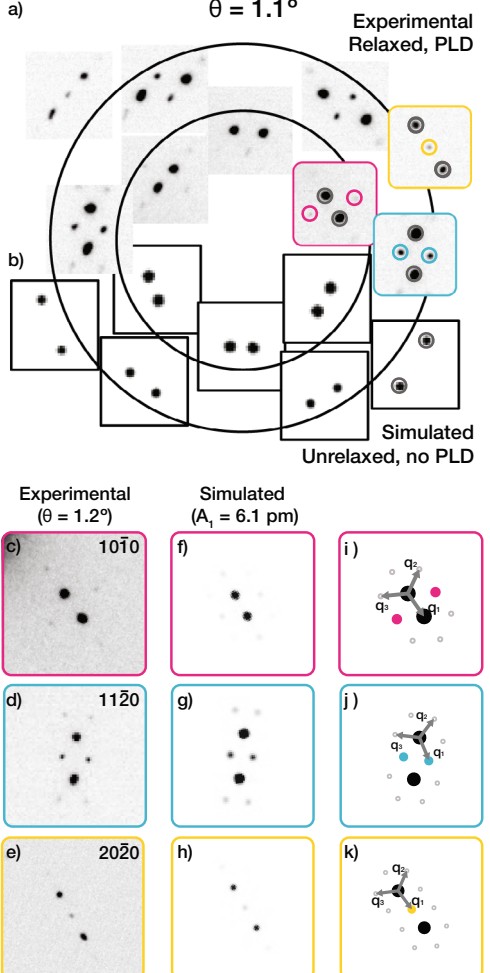

**Fig. 2 | Torsional periodic lattice distortion (PLD) model. a** Electron diffraction of TBG ($\theta = 1.1°$) displays superlattice peak complexes in addition to two sets of Bragg peaks, marked by gray circles. **b** The simulated diffraction pattern of unrelaxed TBG only shows two sets of Bragg peaks. Multislice simulation (**f**–**h**) for PLD with only a single harmonic ($A_1 = 6.1$ pm) matches greatly with the experimental data (**c**–**e**). **i**–**k** Schematic illustration shows PLD wave vectors ($q_i$'s) in relation to Bragg and superlattice peaks.

that decorate Bragg peak pairs and appear more pronounced around higher-order Bragg peaks. The superlattice peaks represent a symmetry reduction beyond that from the global twist angle. The torsional PLD superlattice peaks in TBG at 1.1° are shown (Fig. 2a). The azimuthal intensity distribution of superlattice peaks in SAED of 1.1° TBG implies transversity of the distortion wave ($\hat{\mathbf{A}} \perp \mathbf{q}$).

### Diffraction of Moiré materials

PLDs diffract into reciprocal space as superlattice peaks that surround each Bragg peak. These superlattice peaks are positioned $\alpha\mathbf{q}$ away from Bragg peaks (Fig. 2i–k). The superlattice peaks have intensities proportional to $|J_\alpha(\mathbf{k} \cdot \mathbf{A})|^2$ where $J_\alpha$ is a Bessel function of the first kind, $\mathbf{k}$ is the position of the superlattice peak in reciprocal space, and $\alpha$ is an integer[3,22,25]. For typical values of $\mathbf{k}$ and $\mathbf{A}$, the Bessel function monotonically increases with $|\mathbf{k}|$ and decreases inversely with the integer $\alpha$. The appearance of strong superlattice peaks at high-order Bragg spots (i.e., at larger $|\mathbf{k}|$) is a signature of periodic lattice distortions (PLD) (Supplementary Fig. S1). An analytic expression for the diffraction of twisted 2D materials with a torsional PLD is provided in the Methods.

The dot product ($\mathbf{k} \cdot \mathbf{A}$) reveals the transversity of PLDs in twisted bilayer materials. In reciprocal space, the transverse PLDs manifest as a distribution of superlattice peaks that are stronger along the azimuthal

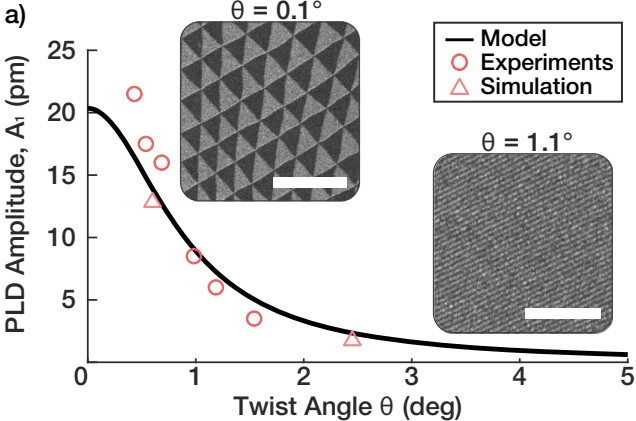

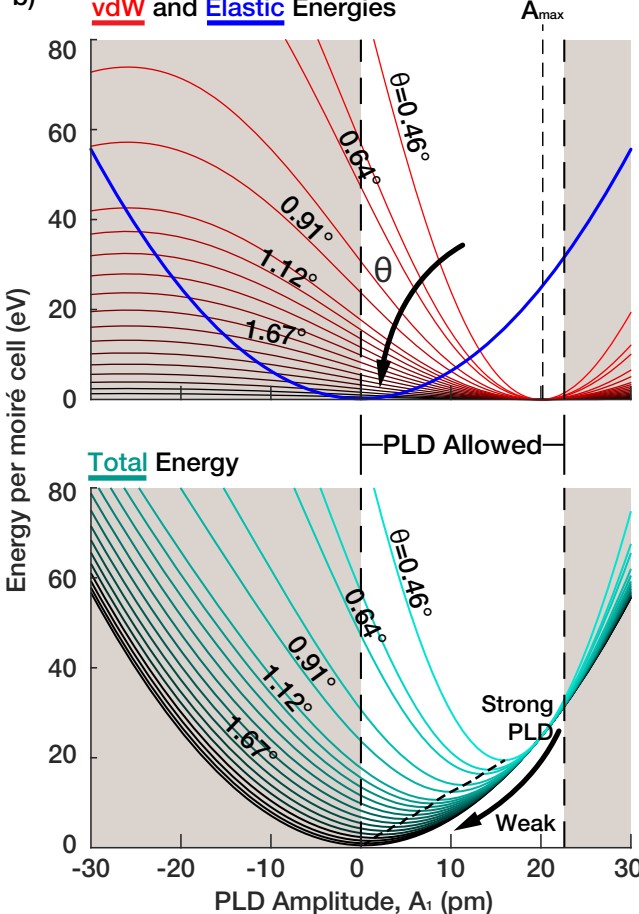

**Fig. 3 | Energy and amplitude of PLDs in twisted bilayer graphene (TBG). a** Torsional PLD model predicts the PLD amplitude ($A_1$) of TBG vs twist angle ($\theta$) follows Lorentzian. Extracted torsional PLD amplitude ($A_1$) from experimental SAED (red circles) and computationally relaxed (red triangles) matches well with the Lorentzian model. Scale bars are 500 nm (0.1°) and 150 nm (1.1°). **b** Top panel: $V_{vdW}$ $V_{vdW}$ (red) and $V_{El}$ $V_{El}$ (blue) of TBG with torsional PLD. The lighter region denotes geometrically-allowed PLD amplitude ($0 \leq A_1 \leq 22.6$ pm). Note that the elastic cost of torsional PLD is twist angle independent, and $V_{vdW}$ $V_{vdW}$ is proportional to $\theta^{-2}$ and has a minimum at $A_1 = 20.35$ pm. Bottom panel: total energy landscape ($V_{vdW} + V_{El}$) ($V_{vdW} + V_{El}$) of TBG with torsional PLD. $\theta$ dependence of $V_{vdW}$ shifts the total energy minimum to stronger PLD.

direction (Supplementary Fig. S1c). In contrast, a longitudinal PLD would produce superlattice peaks that become stronger radially along Bragg vectors (Supplementary Fig. S1b). The torsional PLD in TBG results from the superposition of three transverse PLDs.

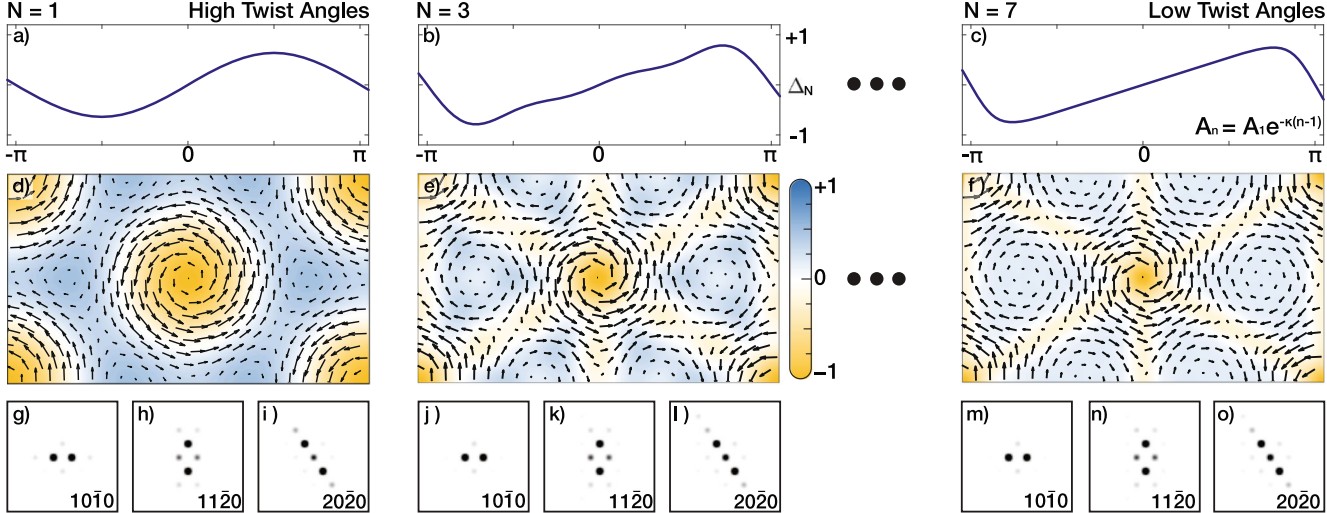

**Fig. 4 | PLDs as a Fourier series in 2D Moiré materials. a–c** Evolution of periodic wave (Δ) as higher harmonic waves **b** $N = 3$, **c** $N = 7$ are included; Δ is normalized displacement magnitude over one period along one direction. Fourier coefficients ($A_n$) are tailored as exponential decay, which produces to a smooth "sawtooth"-like waveform. Including harmonic waves allows high-frequency (i.e., sharp) features in resultant waves. **d–f** Torsional PLD structure with higher harmonic PLD included. The color denotes the amount of local rotation ($\Omega_N$) due to the PLD displacement field (arrows). **g–i** Quantum mechanical electron diffraction simulations of TBG with single harmonic torsional PLD capture the distortions in high twist angles (i.e., near magic angle and higher) well. **j–o** Adding higher harmonics slightly modifies the diffraction patterns and shows better matches with low twist angle systems.

The torsional PLD quantitatively describes experimental observations of twisted bilayer diffraction—superlattice peaks near higher-order Bragg peaks have higher intensities, and superlattice peak intensities increase monotonically as PLD amplitude increases. The torsional PLD in TBG is validated by quantum mechanical multislice simulation. Simulated SAED patterns (Fig. 2f–h) show excellent agreement with experimental data (Fig. 2c–e). More specifically, the relative superlattice to Bragg peak intensity and distribution of simulated superlattice peaks are consistent.

The torsional PLD in TBG is primarily described by a single amplitude coefficient ($A_1$). We report a torsional PLD amplitude ($A_1$) of $7.8 \pm 0.6$ pm for $\theta = 1.1°$ and $6.1 \pm 0.4$ pm for 1.2° near the magic angle in TBG. The PLD amplitude was quantified by matching experimental and simulated diffraction intensities (See Supplementary Fig. S2). Torsional PLD amplitudes for additional twist angles are plotted in Fig. 3a—showing a decrease in amplitude as the twist angle is increased and ultimately disappearing above 3.89°.

### Critical twist angles and the low-twist regime

Below a 3.89° twist angle, atomic distortions in TBG exceed a picometer ($A_1 > 1$ pm). The periodic relaxation of TBG results from competition between interlayer van der Waals stacking energy benefit ($V_{vdW}$) and elastic cost of distortion ($V_{El}$)[26]. Elastic energy cost to accommodate a torsional PLD with amplitude $A_1$, assuming Hookean elasticity, is $2\sqrt{3}\pi^2 G A_1^2$ for each layer, where $G$ is the shear modulus of graphene (9.01 eV/Å$^2$ [20]) (Fig. 3b, blue, derived in Supplementary Note 3). Notably, the elastic energy per supercell is independent of the twist angle, and hence of moiré supercell size. For van der Waals interaction, $V_{vdW}$, we employ Kolmogorov and Crespi's model[27] and compute the interlayer energy as a function of PLD amplitude (See Methods). $V_{vdW}$ has two salient features: first, energy per moiré supercell is proportional to the area of the cell ($\propto \theta^{-2}$) and second, $V_{vdW}$ minimum is at $A_1 = A_{max} = 20.35$ pm (Fig. 3b, top red). Therefore, at large $\theta$ where $V_{El}$ dominates total energy, the total energy is at minimum at small $A_1$. In contrast, as $\theta$ decreases, $A_1$ approaches 20 pm.

We report an upper and lower bound of the PLD amplitude, $A_1$, to be 0 and 22.6 pm, respectively. Local rotation due to the torsional PLD ($\Omega_1 = \frac{1}{2}\nabla \times \Delta_1$) near AB region is $\frac{3qA_1}{4}$. For graphene (a = 2.46 Å), $A_1$ of 22.6 pm will restore all local twist in each layer ($|\Omega_1| = \frac{\theta}{2}$). Negative $A_1$ amplitude is energetically unfavorable as it increases the local twist angle, which decreases the AB domain size. See Supplementary Note 2 for a detailed discussion.

The interlayer interaction energy $V_{vdW}$ of TBG is excellently approximated by a quadratic function within the geometrically-allowed region of $A_1$ ($0 \le \theta \le 22.6$ pm). A non-linear least squares fit gives a semi-empirical model of $V_{vdW} = \frac{2v_0}{\theta^2}(A_1 - A_{max})^2$ where $v_0 = 0.0732$ eV Å$^{-2}$ and $A_{max} = 20.35$ pm. Notably, $A_{max}$ corresponds to energetically allowed maximum $A_1$. The total energy of TBG with a torsional PLD is $V_{tot} = 4\sqrt{3}\pi^2 G A_1^2 + \frac{2v_0}{\theta^2}(A_1 - A_{max})^2$. Minimizing $V_{tot}$ with respect to $A_1$ gives a Lorentzian function:

$$A_1(\theta) = A_{max}\left(1 + \frac{2\pi^2\sqrt{3}G}{v_0}\theta^2\right)^{-1} \tag{2}$$

Eq. (2) (Fig. 3a, black) matches excellently with $A_1$ extracted from experimental data and simulations (Fig. 3a red). The amplitude of the PLD exceeds 1 pm below 3.89° twist—which we define as the low-twist angle regime in TBG. Note, only at the lowest angles below -0.5° do we see a slight underestimation from the Lorentzian model; suggesting higher-order distortions become noticeable.

### Sharp PLD boundaries at extreme low-twist angles

We report a lower twist angle ($\theta \lesssim 0.5°$), TBG relaxes with more complexity, thus roughly defining the extreme low-twist regime. Comparing diffraction patterns at higher $\theta$ (e.g., 1.1°, Fig. 2a) and lower $\theta$ (e.g., 0.4°, Fig. 5a), lower twist angle SAED patterns show not only stronger superlattice peaks, but also different distribution of superlattice peaks with higher-order superlattice peaks. This is attributed to the sharpening of soliton boundaries between AB and BA domains. Yoo et al.'s work suggested that dislocation boundaries become more well-defined at low twist angles using DF-TEM. Even at zero-twist, soliton boundaries have been reported[13,17]. In the extreme low-$\theta$ regime, shear soliton boundaries reach a minimum width previously reported to be $6.2 \pm 0.6$ nm[13]. However, even when soliton boundaries have minimal width, as $\theta$ decreases, these boundaries become a smaller fractional area of the moiré supercell. Thus, at extremely low-twist the PLD needs

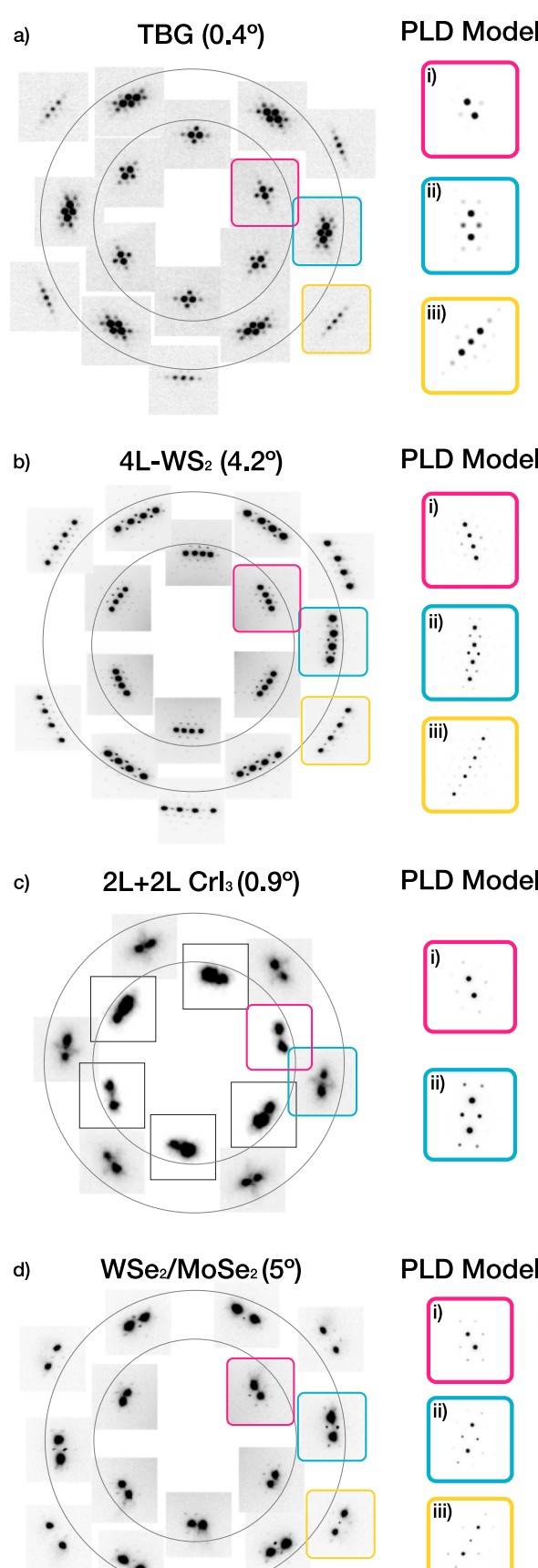

**Fig. 5 | Universal torsional PLD relaxation of twisted 2D materials.** Periodic relaxation is observed universally in multiple twisted 2D systems. SAED of **a** low-$\theta$ TBG, **b** twisted four-layer (4L) $WS_2$ homostructure, **c** twisted bilayer (2L + 2L) $CrI_3$, **d** twisted $WSe_2/MoSe_2$ heterostructure shows bright Bragg peaks with small superlattice peaks. Insets i–iii are multislice simulated diffraction patterns with a torsional PLD model. The torsional PLD model reproduces qualitatively accurate SAED patterns across multiple systems.

to be generalized to include an additional number (N) of Fourier harmonics to accommodate sharper boundaries:

$$\Delta_N = \sum_{n=1}^{N} \Delta_n \qquad (3)$$

The Fourier coefficient $A_n$ dictates the texture of a torsional PLD. Figure 4d–f shows the evolution of a torsional PLD as higher-order Fourier harmonics are included. The arrows represent the displacement field of the torsional PLD and the colored overlay represents the local rotational field ($\Omega_N$, see Supplementary Note 3) in one of the layers; the opposing layer has an equal and opposite local rotational field ($-\Omega_N$). Figure 4d shows a torsional PLD with a single coefficient. The PLD rotational field reveals most of the relaxation is facilitated through twisting circular AA regions (orange). In contrast, with higher harmonics included (Fig. 4e, f) triangular AB/BA regions are anti-twisted (blue) to maximize Bernal stacking within the system in addition to twisting AA regions. Fourier coefficient design produces $\Omega_N$ field pattern that matches previously reported experimental results of local twist fields[18]. Although each harmonic PLD wave contributes elastic energy independently (see Supplementary Note 3.2), this is not true for the interlayer van der Waals energy. In Fig. 4e, f, $A_n$ decays exponentially ($A_n = A_1 e^{-\kappa(n-1)}$); analogous to a smooth 'sawtooth'-like wave in a one-dimensional wave (Fig. 4a–c). Notably, for a smooth—i.e., infinitely differentiable—wave, exponential decay is the upper bound for Fourier coefficients (Paley–Wiener theorem).

The PLD amplitudes for higher harmonics ($A_n$) are calculated by minimizing the total interlayer and intralayer energies (See Methods). Each higher-order term becomes non-negligible incrementally at smaller angles. $A_2$, $A_3$, and $A_4$ will exceed 1 pm at $\theta \lesssim 0.9°$, 0.45°, and 0.3°, respectively (See Supplementary Fig. S3). We describe the extreme low-twist regime to be when $A_3$ becomes significant (>1 pm), however, this demarcation is imprecise. $A_n$ decays exponentially with coefficient $\kappa(\theta)$ linearly proportional to the twist angle ($A_n = A_1 e^{-\kappa(\theta)\cdot(n-1)}$). Thus, decreasing $\theta$ retards decay of $A_n$ and higher harmonics becomes more significant (See Supplementary Fig. S4). The fundamental PLD amplitude, $A_1(\theta)$, remains well described by the empirical Lorentzian, Eq. (2), even when higher-order harmonics are present. If harmonics are ignored, the computed value of $A_1$ will deviate by more than 1 pm at twist angles below $\theta < 0.25°$ but never exceed ~10%. Noticeably, the inclusion of higher harmonics creates a torsional PLD texture that allows enhancement of $A_1$ in the extreme low twist angle regime.

Multislice simulation of SAED (Fig. 4g–o) shows that the distribution of superlattice peaks changes with the presence of higher harmonics. $n^{th}$ harmonic PLD waves add intensity to superlattice peaks $n\alpha\mathbf{q}$ away from each Bragg peak ($\alpha$ is the integer in $J_\alpha$). Figure 4g–o shows simulated TBG diffraction patterns with higher harmonics have stronger superlattice peaks further away from the Bragg peaks. The change is subtle because higher-order harmonics are exponentially weaker.

As $\theta$ nears zero, many higher harmonics (N) are needed and the Fourier basis is more cumbersome. Instead, a hard-domain model, where the superlattice is treated as a quilt of AA, AB, and BA domains with dislocation boundaries, may also become suitable. In the limit of zero angle twist, boundaries become the stacking fault boundaries reported by Brown et al. for untwisted bilayer graphene[17].

## Torsional PLDs across many twisted 2D materials

Torsional PLDs in twisted 2D materials are a universal phenomenon and not limited to TBG[8–10,28]. Figure 5 shows SAED patterns that exhibit periodic relaxations of four distinct twisted 2D systems: (a) low twist angle TBG, (b) four-layer of $WS_2$ (4L-$WS_2$), (c) twisted double bilayer $CrI_3$ (2L + 2L $CrI_3$), and (d) twisted $WSe_2/MoSe_2$ heterostructure.

The relaxation behavior is present in layered transitional metal dichalcogenides ($MX_2$) and trihalides ($MH_3$). Figure 5b shows the diffraction pattern of the four-layer homostructure of $WS_2$ with equal twist angles between layers. Surprisingly, a strong torsional PLD is observed, despite having a large twist angle ($\theta \approx 4°$)[10]. For a multi-layered system, relaxation may not be equivalent between layers. For 4L-$WS_2$, for example, the PLD amplitudes are strongest for the inner-most layers. Here equal and opposite PLDs of the inner two layers matches simulated diffraction patterns (Fig. 5b).

Figure 5c shows four layers of twisted $CrI_3$, a magnetic 2D material, but with a twist only between the middle two layers[9]. Xie et al. reported that this system shows magnetic behavior that cannot be explained by either two-layer or four-layer $CrI_3$ system and periodic relaxation must be accounted for to fully explain the materials properties. For the 2L + 2L $CrI_3$ system, the magnetic properties suggest that the outer layers are distorted together with the inner layers (i.e., each bilayer acts like a monolayer). This is also consistent with diffraction simulations (Fig. 5c).

Periodic reconstruction of twisted materials is not limited to homostructures. $WSe_2/MoSe_2$ heterostructures exhibit twist angle-dependent excitonic behavior[29]. In Fig. 5d, we reveal that the heterostructure with $\theta \approx 5°$ periodically relaxes, despite having different lattice constants. A torsional PLD model is also applicable to such heterostructures. Simulated diffraction patterns (Fig. 5d i–iii) show good agreement with the experimental diffraction pattern. It should be noted that the relaxation behavior of non-graphitic systems will be different when the stacking energy landscape is distinct[21]. Any stacking energy landscape can be accommodated by assigning the appropriate phase to each of the three PLD waves in a torsional PLD (See Supplementary Fig. S8 for more detail). Furthermore, due to lattice constant mismatch, reconstruction of heterobilayer involves compression and expansion of constituent layers[30,31]. Compression/expansion of heterobilayer can be easily incorporated by including longitudinal components to $\hat{\mathbf{A}}_i$'s as demonstrated in Supplementary Fig. S10.

## Discussion

Twisted 2D materials are complex moiré patterns where crystals deform according to a competition between intralayer elastic strain and interlayer van der Waals interactions. A reduction of symmetry arises not only from the global twist between layers but also from the subsequent lattice restructuring. Thus, the twist angle alone provides an incomplete description of the system. We show a torsional periodic lattice distortion is a precise order parameter to describe the atomic structure of twisted materials. Torsional periodic lattice distortions are comprised of three transverse PLDs that maximize the lower energy stacked domains and minimize and form solitonic shear boundaries in between. The amplitude and wave vector of torsional PLDs are defined by the twist angle and, in TBG, can be analytically and empirically predicted with picometer precision. In this sense, moiré materials are PLD engineering at low twist angles.

Despite the real-space complexity of low-twist moiré materials, the entire structure is sparsely described by a single value: the amplitude of the distortion wave. This choice of basis re-frames our understanding of low-twist angle materials. In the case of twisted bilayer graphene, the torsional PLD amplitude can be analytically calculated from the twist angle alone. Although the amplitude of the PLD can change gradually, the overall symmetry reduction occurs instantaneously—therefore, a continuous phase transition is not expected. Although this work thoroughly describes TBG, it is extendable to a variety of 2D materials and twist angles—each with a bespoke set of PLDs to match the interlayer energy landscapes.

## Methods

### Electron diffraction of torsional PLD

In the presence of a single torsional PLD with three wave vectors ($\mathbf{q}_1$, $\mathbf{q}_2$, $\mathbf{q}_3$), the reciprocal structure of the top layer ($V_{top}(\mathbf{k})$) is described by:

$$
\begin{aligned}
V_{top}(\mathbf{k}) = \sum_{\mathbf{b}} &\delta(\mathbf{k} - \mathbf{b}) J_0(\mathbf{k} \cdot \mathbf{A}_1) J_0(\mathbf{k} \cdot \mathbf{A}_2) J_0(\mathbf{k} \cdot \mathbf{A}_3) \\
&\pm \delta(\mathbf{k} - \mathbf{b} \pm \mathbf{q}_1) J_1(\mathbf{k} \cdot \mathbf{A}_1) J_0(\mathbf{k} \cdot \mathbf{A}_2) J_0(\mathbf{k} \cdot \mathbf{A}_3) \\
&\pm \delta(\mathbf{k} - \mathbf{b} \pm \mathbf{q}_2) J_0(\mathbf{k} \cdot \mathbf{A}_1) J_1(\mathbf{k} \cdot \mathbf{A}_2) J_0(\mathbf{k} \cdot \mathbf{A}_3) \\
&\pm \delta(\mathbf{k} - \mathbf{b} \pm \mathbf{q}_3) J_0(\mathbf{k} \cdot \mathbf{A}_1) J_0(\mathbf{k} \cdot \mathbf{A}_2) J_1(\mathbf{k} \cdot \mathbf{A}_3) \\
&+ \mathcal{O}((kA)^2)
\end{aligned}
\tag{4}
$$

The bottom layer's reciprocal structure ($V_{bot}(\mathbf{k})$) has the same form with $V_{top}(\mathbf{k})$ but with equal and opposite PLD amplitude ($\mathbf{A}_i^{top} = -\mathbf{A}_i^{bot}$). The measured diffraction pattern is the squared magnitude $|V_{top}(\mathbf{k}) + V_{bot}(\mathbf{k})|^2$. The first term represents Bragg peaks. The second term is first-order superlattice peak. Here $\mathbf{k} \cdot \mathbf{A}$ is small (less than one) and the higher-order terms $\mathcal{O}((kA)^2)$ are omitted for simplicity.

### Transmission electron microscopy and diffraction

JEOL 2010F operated at 80 keV with Gatan OneView Camera was used for SAED and DF-TEM imaging of TBG. 4L-WS2 and 2L + 2L $CrI_3$ SAEDs were taken on Thermo Fisher Talos operated at 200 keV with Gatan OneView Camera. $WSe_2/MoSe_2$ heterostructure SAEDs were taken on JEOL 3100R05 operated at 300 keV. AB/BA domain contrast on DF-TEM was obtained by placing an objective aperture on $\langle 10\bar{1}0 \rangle$ with specimen as demonstrated previously by refs. 6, 17.

Bragg and superlattice peak intensities were quantified by non-linear least squares fitting six-parameter 2D Gaussian peaks and calculating the volume under each fitted Gaussian.

### Electron diffraction simulation

Heavy atoms in twisted 2D materials can have a small but non-negligible influence on the quantification of superlattice peaks, especially for low-incident electron energies. Quantum mechanical multi-slice simulations can provide better quantification[32]. However, the magnitude and distribution of intensities make the presence of torsional PLD immediately obvious.

Fully quantum mechanical multislice simulations are performed to match experimental SAEDs. Multislice algorithm simulates dynamic scatterings of swift electrons by slicing specimens into multiple, thin slices. E. J. Kirkland's software (autoslic)[33] with matching incident electron conditions with experiments was used to calculate. A 300 Å radius disk-shaped TBG crystal was placed on $1200 \times 1200$ Å$^2$ area to reduce wraparound artifact. The multislice algorithm was set to slice crystal every 0.5 Å. Electron wavefunctions were sampled at $4096 \times 4096$ pixels. Simulation parameters for Fig. 5b–d were similar. Simulations were averaged over 16 frozen phonon configurations to simulate thermal diffused scattering. See https://doi.org/10.6084/m9.figshare.20352933.v1 for simulation parameters.

### Twisted bilayer graphene energy calculations

vdW interlayer registry energies were calculated using Kolmogorov−Crespi potential with known lattice parameters for graphitic systems ($a = 2.46$ Å, $c = 3.34$ Å) and assumed both graphene layers are perfectly flat. The energy at each atomic site was calculated by summing overall interactions with atoms in the opposing layer. $V_{vdW}$ was calculated by integrating registry energies over a moiré unit cell. The calculation was done over many twist angles (0.463°–6.009°) with up to 61,348 atoms per layer.

Elastic energies ($V_{El}$) were analytically derived from elasticity tensor assuming plane stress deformation and using previously reported shear modulus value ($G = 9.01$ eV/$Å^2$ [20]). See Supplementary Note 3 for detailed derivations.

Higher harmonic PLD amplitudes ($A_n$) were calculated by minimizing the total energy using the simplex algorithm implemented in MATLAB (fminsearch).

Calculation of computationally relaxed structures (Fig. 3a red triangles) was done by a multiscale computational method described in detail in ref. 20.

## Sample preparations

**Twisted bilayer graphene.** Graphene and h-BN are mechanically exfoliated onto Si/SiO₂ (285 nm) substrates. Monolayer graphene was identified by optical contrast and Raman spectroscopy. After preparing graphene and h-BN flakes, the thin single crystallite h-BN layer was first picked up at 70 °C using poly(Bisphenol A carbonate) coated on a polydimethylsiloxane stamp. Then, the h-BN layer was engaged to half of the graphene flake. By lifting the stamp off the substrates, we pick up only the part of the graphene which was covered by the top h-BN layer, leaving the remaining part of the graphene on the substrate. After the detachment, the substrate was rotated at a controlled angle. Engaging the graphene/h-BN stack on the adhesive polymer to the other half piece of graphene on the substrate makes the artificial bilayer graphene with a controlled twist angle. The whole stack was then transferred onto a thin SiN membrane TEM grid.

**4L-WS₂.** Large monolayer WS₂ single crystals were synthesized by metal-organic chemical vapor deposition (MOCVD) on 300 nm oxide silicon wafers[34]. A single triangular domain from the growth was selected, then tear-and-stacked four times with a twist of 4 degrees and delaminated onto a Si/SiO₂ substrate by a vacuum assembly robot[10]. To prepare the twisted four-layer WS₂ for TEM imaging, the sample was spun coated with PMMA and floated in 1 M KOH until the substrate detached from the PMMA, then wet transferred onto a 1 μm holey carbon/copper TEM grid. Lastly, the TEM grid was solvent cleaned in acetone to remove all polymer (from stamp residue and PMMA).

**2L+2L CrI₃.** For the fabrication of twist double bilayer (tDB) CrI₃, the bilayer CrI₃ flakes were firstly exfoliated and identified inside a nitrogen gas-filled glovebox and assembled with a desirable twist angle by the 'tear-and-stack` method. tDB CrI₃ samples were double encapsulated with thin hexagonal boron nitride (h-BN) flakes (~5 nm) to prevent the degradation from the oxygen and moisture from the ambient condition. The prepared tDB CrI₃ samples were finally transferred onto the TEM grids, cleaned with chloroform solvent, and dried in nitrogen gas.

**WSe₂/MoSe₂ heterobilayer.** Monolayers of TMDs were prepared via mechanical exfoliation from bulk crystals (from HQ Graphene) onto a PDMS stamp (PF Film X0 from Gel-Pak). The PDMS stamp was pre-treated and treated by oxygen plasma to minimize the PDMS residue left on the monolayers. Heterostructures were fabricated using the PDMS-assisted dry transfer technique operating under a home-built setup with micromanipulators, and a rotational stage, and the mono-layers were aligned under an optical microscope. Heterostructures were annealed in a vacuum at 130 °C for 1 h.

## Data availability

Raw SAED micrographs are publicly available in (https://doi.org/10.6084/m9.figshare.20352933.v2). Any additional data that support the findings of this study are available in the article and Supplementary Information.

## Code availability

The simulation parameters for running multislice simulation are publicly available (https://doi.org/10.6084/m9.figshare.20352933.v2).

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

## Acknowledgements

R.H. acknowledges support from ARO grant no. W911NF-22-1-0056. S.H.S. acknowledges support from the W.M. Keck Foundation. P.K. acknowledge support from ARO MURI (W911NF-21-2-0147). R.E. acknowledge support from NSF DMR-1922172. L.Z. acknowledges support by NSF CAREER grant No. DMR-174774, AFOSR YIP grant No. FA9550-21-1-0065 and Alfred P. Sloan Foundation. J.P., A.J.M., and A.Y. acknowledge funding from the National Science Foundation Platform for the Accelerated Realization, Analysis, and Discovery of Interface Materials (PARADIM) under Cooperative Agreement No. DMR-2039380 and the Air Force Office of Scientific Research MURI project (FA9550-18-1-0480). A.J.M. was supported by the Kadanoff-Rice Postdoctoral Fellowship of the University of Chicago MRSEC (DMR-2 011854). A.Y. was supported by the Department of Defense (DoD) through the National Defense Science and Engineering Graduate (NDSEG) Fellowship Program. Diffraction data collected by the authors for use in Figs. 2 and 5 have been adapted and reused in part from refs. 6, 9 and 10 as permitted by Springer Nature.

## Author contributions

S.H.S., Y.M.G., H.Y., R.E., P.K., and R.H. performed electron microscopy on twisted materials. H.Y., R.E., and P.K. fabricated TBG. H.X. and L.Z. fabricated 2L + 2L CrI$_3$. K.Z. and E.B.T. calculated relaxed structures for TBG. Z.L. and P.B.D. fabricated WSe$_2$/MoSe$_2$. A.Y., A.J.M., and J.P. fabricated a 4L-WS$_2$ sample. S.H.S., Y.M.G., and R.H. prepared the manuscript. All authors reviewed and edited the manuscript.

## Competing interests

The authors declare no competing interests.
