## [Peer Review File · Nature Communications]

Reviewer Comments, first round -

Reviewer #1 (Remarks to the Author):

Sung et al present a mathematical description of periodic lattice relaxation in twisted 2D heterostructures. The modulation of stacked 2D lattices when twisted to small angles is an important topic in physics and material science with the effect of lattice relaxation having previously shown to modify local band structure. It is also a subject where the modulation has been mostly described in a qualitative rather than a quantitative manner in existing literature. This work also presents the verification of the reconstruction behaviour through the observation of superlattice spots in the diffraction data making it applicable even for materials which are highly electron beam sensitive. The manuscript therefore offers the potential for significant enhancement over existing work.

However, the key description of torsional periodic lattice distortions is verified by comparison of "multislice" diffraction simulations with experimental data. Little information is given regarding how the diffraction spot simulations are achieved and we have been unable to reproduce these results from the methodological information provided in the manuscript. Particularly, the size of the real space supercell necessary for small diffraction angles and the small step size required in reciprocal space to effectively resolve superlattice reflections location and intensity, results in a complex multislice simulation which is hard to achieve using the widely adopted E. Kirkland multislice code [ref 31] and the processing power of a conventional PC.

A further concern is that I do not assess that Figure 5c has sufficient quality to claim the presence of superlattice spots in the experimental data and hence the claim to have verified lattice reconstruction.

Reviewer #2 (Remarks to the Author):

In this paper by Sung and co-workers, a torsional periodic lattice distortion (PLD) model is proposed to describe lattice reconstruction in twisted 2D materials. Theoretical results, including simulated electron diffraction patterns and displacement fields as well as strain and stacking energy calculations are accompanied by experimental selected area diffraction data. The orientations and relative intensities of superlattice peaks observed in diffraction provide evidence of transverse PLDs in moiré superlattices that evolve as a function of twist angle.

Discussion of lattice reconstruction in twisted 2D materials is currently of significant interest due to its impact on the exotic physical behavior observed in these systems. This paper provides an atomic scale description of the reconstruction process, whereas current literature has thus far focused on nanoscale pictures of lattice relaxation. The authors also show how the PLD model explains the changing complexity of reconstruction when transitioning from a low-twist to an extreme low-twist angle regime, which has been previously observed experimentally but not fully understood. Overall, the results from this work build nicely on the existing literature on lattice reconstruction (i.e. DF-TEM, 4D-STEM and HAADF-STEM studies, theoretical predictions) and fill in important gaps in the understanding of how reconstruction occurs in twisted 2D systems. I recommend this paper for publication in Nature Communications once the following comments are addressed:

a. It would be very insightful if more detail were provided regarding the application of the PLD model to the multilayer systems shown in Figures 5b and c. Should one assume that the model described for the bilayer in the Methods section can be directly applied to the 4L system by continuing to alternate the sign of the PLD amplitude? For the 2L + 2L system, what happens in the outermost untwisted layers? Do they pin the distortions of the innermost layers, or do they also undergo PLD?

b. The authors imply that reconstruction in heterobilayers (e.g., WS₂/MoSe₂) can be described by transverse PLDs. However, other reports on WSe₂/WS₂ (1) and hBN/graphene (2) have suggested

that heterobilayer reconstruction involves compression/expansion of the constituent layers, which seems more like a longitudinal PLD. How do the authors rationalize this discrepancy? A more in-depth analysis of the superlattice peaks and atomic displacements in this system would be helpful.

References:

- 1.H. Li et al. Nat. Mater. 2021, 20, 945–950.
- 2.C.R. Woods et al. Nat. Phys. 2014, 10, 451–456

c. The experimental superlattice peaks in Figure 5c and, to a lesser degree, in Figure 5d are very difficult to see and to compare with the peaks in the PLD models. This could be due to disorder in the sample. It would be good if the authors provided DF-TEM images of the areas where SAED was collected to ascertain the level of disorder in the probed areas.

d. The authors should include descriptions for sample preparation in the Methods section.

e. A minor note: The main text says transverse PLDs are shown in Fig. S1b and longitudinal PLDs are in Fig. S1c, but these seem to be swapped in the figure itself.

REVIEWER COMMENTS

Reviewer #1 (Remarks to the Author):

Sung et al present a mathematical description of periodic lattice relaxation in twisted 2D heterostructures. The modulation of stacked 2D lattices when twisted to small angles is an important topic in physics and material science with the effect of lattice relaxation having previously shown to modify local band structure. It is also a subject where the modulation has been mostly described in a qualitative rather than a quantitative manner in existing literature. This work also presents the verification of the reconstruction behaviour through the observation of superlattice spots in the diffraction data making it applicable even for materials which are highly electron beam sensitive. The manuscript therefore offers the potential for significant enhancement over existing work.

We appreciate positive comments and highlighting the importance of a quantitative description of lattice relaxation in twisted 2D materials.

However, the key description of torsional periodic lattice distortions is verified by comparison of “multislice” diffraction simulations with experimental data. Little information is given regarding how the diffraction spot simulations are achieved and we have been unable to reproduce these results from the methodological information provided in the manuscript.

We now include a more detailed description of the simulation methods. The method section on “Electron Diffraction Simulation” on page 8 now reads “A 300 Å radius disk shaped TBG crystal was placed on 1200×1200 Å² area to reduce wraparound artifact. The multislice algorithm was set to slice crystal every 0.5 Å. Electron wavefunctions were sampled at 4096×4096 pixels. Simulation parameters for Figure 5b–d were similar.” We hope this increases reproducibility of this manuscript’s results.

Particularly, the size of the real space supercell necessary for small diffraction angles and the small step size required in reciprocal space to effectively resolve superlattice reflections location and intensity, results in a complex multislice simulation which is hard to achieve using the widely adopted E. Kirkland multislice code [ref 31] and the processing power of a conventional PC.

We are pleased the reviewer has invested their time to replicate our results. To streamline this for the reviewer we now provide two .xyz crystal files of 1.1° twisted bilayer graphene (with and without PLD) and simulation parameters for E. J. Kirkland multislice code. Because the specimen is extremely thin, the simulation requires very few wave propagations to compute even at 0.5 Å slice thickness. For reference, TBG SAED pattern for Fig 5a was computed in about a minute on the authors’ laptop. The simulation parameters are now publicly available (doi: 10.6084/m9.figshare.20352933.v1).

The method section on “Electron Diffraction Simulation” now reads “... See DOI: 10.6084/m9.figshare.20352933.v1 for simulation parameters.).

A further concern is that I do not assess that Figure 5c has sufficient quality to claim the presence of superlattice spots in the experimental data and hence the claim to have verified lattice reconstruction.

To clearly verify the presence of periodic lattice distortions, we now include DF-TEM measurements (added as Supplemental Fig. S9) where the expected array of triangular domain due to PLD restructuring is clearly visible in real-space. The triangular domain is a definitive signature of lattice relaxation [1]. This is also consistent with magnetic property measurement reported in [2].

[1] J. S. Alden et al., PNAS **110** (28) 11256–11260 (2013)

[2] H. Xie et al., Nat. Phys. **18** 30–36 (2022)

Reviewer #2 (Remarks to the Author):

In this paper by Sung and co-workers, a torsional periodic lattice distortion (PLD) model is proposed to describe lattice reconstruction in twisted 2D materials. Theoretical results, including simulated electron diffraction patterns and displacement fields as well as strain and stacking energy calculations are accompanied by experimental selected area diffraction data. The orientations and relative intensities of superlattice peaks observed in diffraction provide evidence of transverse PLDs in moiré superlattices that evolve as a function of twist angle.

Discussion of lattice reconstruction in twisted 2D materials is currently of significant interest due to its impact on the exotic physical behavior observed in these systems. This paper provides an atomic scale description of the reconstruction process, whereas current literature has thus far focused on nanoscale pictures of lattice relaxation. The authors also show how the PLD model explains the changing complexity of reconstruction when transitioning from a low-twist to an extreme low-twist angle regime, which has been previously observed experimentally but not fully understood. Overall, the results from this work build nicely on the existing literature on lattice reconstruction (i.e. DF-TEM, 4D-STEM and HAADF-STEM studies, theoretical predictions) and fill in important gaps in the understanding of how reconstruction occurs in twisted 2D systems. I recommend this paper for publication in Nature Communications once the following comments are addressed:

We appreciate the positive support from the reviewer.

a. It would be very insightful if more detail were provided regarding the application of the PLD model to the multilayer systems shown in Figures 5b and c. Should one assume that the model described for the bilayer in the Methods section can be directly applied to the 4L system by continuing to alternate the sign of the PLD amplitude?

Great question. In multi-layered systems a priori that the relaxation amplitudes in each layer should not be assumed equal amongst all layers. The PLD model coefficients are ultimately dictated by the Hamiltonian of the system. Often times, this means the layers will counter-rotate between adjacent layers.

To clarify this, the main text on page 5 now reads "... For multi-layered system, relaxation may not be equivalent between layers. For 4L-WS₂, for example, the PLD amplitudes are strongest for the inner-most layers. Here equal and opposite PLDs of the inner two layers matches simulated diffraction patterns (Fig. 5b)."

For the 2L + 2L system, what happens in the outermost untwisted layers? Do they pin the distortions of the innermost layers, or do they also undergo PLD?

To clarify this, the main text on page 5 now reads "For 2L+2L CrI₃ system, the magnetic properties suggest that the outer layers distorted together with the inner layers (i.e. each bilayer acts like a monolayer). This is also consistent with diffraction simulations (Fig. 5c)."

b. The authors imply that reconstruction in heterobilayers (e.g., WS₂/MoSe₂) can be described by transverse PLDs. However, other reports on WSe₂/WS₂ (1) and hBN/graphene (2) have suggested that heterobilayer reconstruction involves compression/expansion of the constituent layers, which seems more like a longitudinal PLD. How do the authors rationalize this discrepancy? A more in-depth analysis of the superlattice peaks and atomic displacements in this system would be helpful.

References:

- 1.H. Li et al. Nat. Mater. 2021, 20, 945–950.
- 2.C.R. Woods et al. Nat. Phys. 2014, 10, 451–456

We appreciate the reviewer for raising an important point and bringing valuable references to our attention (now added in manuscript). Compression/expansion is readily incorporated by choosing PLD unit vectors ($\hat{\mathbf{A}}_i$'s) to be non-orthogonal to PLD wave vectors (\mathbf{q}_i 's). Regardless of types of nearly-commensurate mismatch (twist, lattice expansion, or both), lattices relax with soliton-like boundaries [26] from two competing energy wells: quadratic elastic energy centered at 0 PLD amplitude and stacking energy off-centered—just like in Fig 3b,c.

We now make this point more salient in the main text on page 6, “Furthermore, due to lattice constant mismatch, reconstruction of heterobilayer involves compression and expansion of constituent layers [30, 31]. In the PLD description of the reconstruction, compression/expansion of heterobilayer can be incorporated by including longitudinal components to $\hat{\mathbf{A}}_i$ as demonstrated in Supplemental Materials S13.”

Previously, this point was subtly implied on page 2 when we state “Transverse distortions are expected when the lattice constants of both layers are equivalent” We now add on page 2 “(otherwise longitudinal components, $\hat{\mathbf{A}}_i \parallel \mathbf{q}_i$, may be present).”

[26] P. Bak, Rep. Prog. Phys. 1982 **45**, 587–629

[30] H. Li et al. Nat. Mater. 2021, 20, 945–950.

[31] C.R. Woods et al. Nat. Phys. 2014, 10, 451–456

We now also include Supplemental Materials S13 to clearly demonstrate this effect.

c. The experimental superlattice peaks in Figure 5c and, to a lesser degree, in Figure 5d are very difficult to see and to compare with the peaks in the PLD models. This could be due to disorder in the sample. It would be good if the authors provided DF-TEM images of the areas where SAED was collected to ascertain the level of disorder in the probed areas.

We agree (as does Reviewer #1) that the superlattice peaks are difficult (but possible) to see in Fig 5c. To clearly verify the presence of periodic lattice distortions, we now include DF-TEM measurements (added as Supplemental Fig. S9) alongside SAED where the expected array of triangular domain due to PLD restructuring is clearly visible in real-space. The triangular domain is a definitive signature of lattice relaxation [1]. This is consistent with magnetic property measurement reported in [2].

[1] J. S. Alden et al., PNAS **110** (28) 11256–11260 (2013)

[2] H. Xie et al., Nat. Phys. **18** 30–36 (2022)

d. The authors should include descriptions for sample preparation in the Methods section.

We thank the reviewer’s suggestion. Extensive description for TEM sample preparation of twisted bilayer graphene, 4L-WS₂, 2L+2L CrI₃ and WS₂/MoSe₂ heterobilayers are now included in the Methods section. We hope this increases reproducibility of this manuscript’s results.

e. A minor note: The main text says transverse PLDs are shown in Fig. S1b and longitudinal PLDs are in Fig. S1c, but these seem to be swapped in the figure itself.

We greatly appreciate the reviewer noting this typo. The paragraph 2 of page 3 of main text now reads “... azimuthal direction (Supplemental Fig. S1c)... radially along Bragg vectors (Supplemental Fig. S1b)...”

REVIEWER COMMENTS

Reviewer #1 (Remarks to the Author):

Sung et al present a mathematical description of periodic lattice relaxation in twisted 2D heterostructures. The modulation of stacked 2D lattices when twisted to small angles is an important topic in physics and material science with the effect of lattice relaxation having previously shown to modify local band structure. It is also a subject where the modulation has been mostly described in a qualitative rather than a quantitative manner in existing literature. This work also presents the verification of the reconstruction behaviour through the observation of superlattice spots in the diffraction data making it applicable even for materials which are highly electron beam sensitive. The manuscript therefore offers the potential for significant enhancement over existing work.

We appreciate positive comments and highlighting the importance of a quantitative description of lattice relaxation in twisted 2D materials.

However, the key description of torsional periodic lattice distortions is verified by comparison of “multislice” diffraction simulations with experimental data. Little information is given regarding how the diffraction spot simulations are achieved and we have been unable to reproduce these results from the methodological information provided in the manuscript.

We now include a more detailed description of the simulation methods. The method section on “Electron Diffraction Simulation” on page 8 now reads “A 300 Å radius disk shaped TBG crystal was placed on 1200×1200 Å² area to reduce wraparound artifact. The multislice algorithm was set to slice crystal every 0.5 Å. Electron wavefunctions were sampled at 4096×4096 pixels. Simulation parameters for Figure 5b–d were similar.” We hope this increases reproducibility of this manuscript’s results.

Particularly, the size of the real space supercell necessary for small diffraction angles and the small step size required in reciprocal space to effectively resolve superlattice reflections location and intensity, results in a complex multislice simulation which is hard to achieve using the widely adopted E. Kirkland multislice code [ref 31] and the processing power of a conventional PC.

We are pleased the reviewer has invested their time to replicate our results. To streamline this for the reviewer we now provide two .xyz crystal files of 1.1° twisted bilayer graphene (with and without PLD) and simulation parameters for E. J. Kirkland multislice code. Because the specimen is extremely thin, the simulation requires very few wave propagations to compute even at 0.5 Å slice thickness. For reference, TBG SAED pattern for Fig 5a was computed in about a minute on the authors’ laptop. The simulation parameters are now publicly available (doi: 10.6084/m9.figshare.20352933.v1).

The method section on “Electron Diffraction Simulation” now reads “... See DOI: 10.6084/m9.figshare.20352933.v1 for simulation parameters.).

A further concern is that I do not assess that Figure 5c has sufficient quality to claim the presence of superlattice spots in the experimental data and hence the claim to have verified lattice reconstruction.

To clearly verify the presence of periodic lattice distortions, we now include DF-TEM measurements (added as Supplemental Fig. S9) where the expected array of triangular domain due to PLD restructuring is clearly visible in real-space. The triangular domain is a definitive signature of lattice relaxation [1]. This is also consistent with magnetic property measurement reported in [2].

[1] J. S. Alden et al., PNAS **110** (28) 11256–11260 (2013)

[2] H. Xie et al., Nat. Phys. **18** 30–36 (2022)

Reviewer #2 (Remarks to the Author):

In this paper by Sung and co-workers, a torsional periodic lattice distortion (PLD) model is proposed to describe lattice reconstruction in twisted 2D materials. Theoretical results, including simulated electron diffraction patterns and displacement fields as well as strain and stacking energy calculations are accompanied by experimental selected area diffraction data. The orientations and relative intensities of superlattice peaks observed in diffraction provide evidence of transverse PLDs in moiré superlattices that evolve as a function of twist angle.

Discussion of lattice reconstruction in twisted 2D materials is currently of significant interest due to its impact on the exotic physical behavior observed in these systems. This paper provides an atomic scale description of the reconstruction process, whereas current literature has thus far focused on nanoscale pictures of lattice relaxation. The authors also show how the PLD model explains the changing complexity of reconstruction when transitioning from a low-twist to an extreme low-twist angle regime, which has been previously observed experimentally but not fully understood. Overall, the results from this work build nicely on the existing literature on lattice reconstruction (i.e. DF-TEM, 4D-STEM and HAADF-STEM studies, theoretical predictions) and fill in important gaps in the understanding of how reconstruction occurs in twisted 2D systems. I recommend this paper for publication in Nature Communications once the following comments are addressed:

We appreciate the positive support from the reviewer.

a. It would be very insightful if more detail were provided regarding the application of the PLD model to the multilayer systems shown in Figures 5b and c. Should one assume that the model described for the bilayer in the Methods section can be directly applied to the 4L system by continuing to alternate the sign of the PLD amplitude?

Great question. In multi-layered systems a priori that the relaxation amplitudes in each layer should not be assumed equal amongst all layers. The PLD model coefficients are ultimately dictated by the Hamiltonian of the system. Often times, this means the layers will counter-rotate between adjacent layers.

To clarify this, the main text on page 5 now reads "... For multi-layered system, relaxation may not be equivalent between layers. For 4L-WS₂, for example, the PLD amplitudes are strongest for the inner-most layers. Here equal and opposite PLDs of the inner two layers matches simulated diffraction patterns (Fig. 5b)."

For the 2L + 2L system, what happens in the outermost untwisted layers? Do they pin the distortions of the innermost layers, or do they also undergo PLD?

To clarify this, the main text on page 5 now reads "For 2L+2L CrI₃ system, the magnetic properties suggest that the outer layers distorted together with the inner layers (i.e. each bilayer acts like a monolayer). This is also consistent with diffraction simulations (Fig. 5c)."

b. The authors imply that reconstruction in heterobilayers (e.g., WS₂/MoSe₂) can be described by transverse PLDs. However, other reports on WSe₂/WS₂ (1) and hBN/graphene (2) have suggested that heterobilayer reconstruction involves compression/expansion of the constituent layers, which seems more like a longitudinal PLD. How do the authors rationalize this discrepancy? A more in-depth analysis of the superlattice peaks and atomic displacements in this system would be helpful.

References:

- 1.H. Li et al. Nat. Mater. 2021, 20, 945–950.
- 2.C.R. Woods et al. Nat. Phys. 2014, 10, 451–456

We appreciate the reviewer for raising an important point and bringing valuable references to our attention (now added in manuscript). Compression/expansion is readily incorporated by choosing PLD unit vectors ($\hat{\mathbf{A}}_i$'s) to be non-orthogonal to PLD wave vectors (\mathbf{q}_i 's). Regardless of types of nearly-commensurate mismatch (twist, lattice expansion, or both), lattices relax with soliton-like boundaries [26] from two competing energy wells: quadratic elastic energy centered at 0 PLD amplitude and stacking energy off-centered—just like in Fig 3b,c.

We now make this point more salient in the main text on page 6, “Furthermore, due to lattice constant mismatch, reconstruction of heterobilayer involves compression and expansion of constituent layers [30, 31]. In the PLD description of the reconstruction, compression/expansion of heterobilayer can be incorporated by including longitudinal components to $\hat{\mathbf{A}}_i$ as demonstrated in Supplemental Materials S13.”

Previously, this point was subtly implied on page 2 when we state “Transverse distortions are expected when the lattice constants of both layers are equivalent” We now add on page 2 “(otherwise longitudinal components, $\hat{\mathbf{A}}_i \parallel \mathbf{q}_i$, may be present).”

[26] P. Bak, Rep. Prog. Phys. 1982 **45**, 587–629

[30] H. Li et al. Nat. Mater. 2021, 20, 945–950.

[31] C.R. Woods et al. Nat. Phys. 2014, 10, 451–456

We now also include Supplemental Materials S13 to clearly demonstrate this effect.

c. The experimental superlattice peaks in Figure 5c and, to a lesser degree, in Figure 5d are very difficult to see and to compare with the peaks in the PLD models. This could be due to disorder in the sample. It would be good if the authors provided DF-TEM images of the areas where SAED was collected to ascertain the level of disorder in the probed areas.

We agree (as does Reviewer #1) that the superlattice peaks are difficult (but possible) to see in Fig 5c. To clearly verify the presence of periodic lattice distortions, we now include DF-TEM measurements (added as Supplemental Fig. S9) alongside SAED where the expected array of triangular domain due to PLD restructuring is clearly visible in real-space. The triangular domain is a definitive signature of lattice relaxation [1]. This is consistent with magnetic property measurement reported in [2].

[1] J. S. Alden et al., PNAS **110** (28) 11256–11260 (2013)

[2] H. Xie et al., Nat. Phys. **18** 30–36 (2022)

d. The authors should include descriptions for sample preparation in the Methods section.

We thank the reviewer’s suggestion. Extensive description for TEM sample preparation of twisted bilayer graphene, 4L-WS₂, 2L+2L CrI₃ and WS₂/MoSe₂ heterobilayers are now included in the Methods section. We hope this increases reproducibility of this manuscript’s results.

e. A minor note: The main text says transverse PLDs are shown in Fig. S1b and longitudinal PLDs are in Fig. S1c, but these seem to be swapped in the figure itself.

We greatly appreciate the reviewer noting this typo. The paragraph 2 of page 3 of main text now reads “... azimuthal direction (Supplemental Fig. S1c)... radially along Bragg vectors (Supplemental Fig. S1b)...”

Reviewer Comments, second round -

Reviewer #2 (Remarks to the Author):

The authors have modified the manuscript to address some of my previous comments. However, there are a couple of remaining issues that I think should be addressed.

1.Regarding the 4L-WS2 system in Fig. 5b: the authors have clarified that there is unequal relaxation in the innermost vs outermost layers of the system. My understanding is that the relaxation in the outer layers has been ignored for the diffraction simulations. While it makes sense to me that the PLD amplitude in the outer layers is weaker, I am not convinced that the PLD in these layers can simply be ignored. For example, there are additional faint peaks present in the experimental data that are not present in the simulations. The authors should provide simulated diffraction data that includes PLD in the outermost layers to confirm whether additional superlattice peaks arise from the relaxation of those layers.

2.Regarding the WS2/MoSe2 system in Fig. 5d: the authors have confirmed longitudinal components are present in heterobilayers. However, I still do not understand the superlattice and Bragg peak orientations. Based on SI Fig. S1b, longitudinal PLDs produce superlattice peaks that extend radially from the primary Bragg peaks, but this does not match what is observed in Fig. 5d. It is also odd that the WS2 and MoSe2 appear to have nearly identical lattice constants when these materials have a relatively large lattice mismatch (~4%). Have the experimental diffraction peaks been rotated at all from their original orientations?

If they have been rotated, then this would answer my questions but should be mentioned. If they have not been rotated, then 1) why are the superlattice peaks oriented azimuthally instead of radially with respect to the primary Bragg peaks? Is this because the twist angle is so large? And 2) why do the WS2 and MoSe2 Bragg peaks have approximately the same radial distance?

Reviewer #3 (Remarks to the Author):

Dear Authors,

Thank you for your responses to the comments and queries raised about the original manuscript submission. I am pleased to see a more complete description of the simulation parameters used and a deeper analysis of some of the other twisted 2D materials presented.

I am happy that the original review comments have been addressed satisfactorily and I am happy to approve the revised manuscript for publication

REVIEWER COMMENTS

Reviewer #1, 3 (Remarks to the Author):

Dear Authors,

Thank you for your responses to the comments and queries raised about the original manuscript submission. I am pleased to see a more complete description of the simulation parameters used and a deeper analysis of some of the other twisted 2D materials presented.

I am happy that the original review comments have been addressed satisfactorily and I am happy to approve the revised manuscript for publication

We appreciate the response and glad to hear our revisions are well received.

Reviewer #2:
Remarks to authors:

The authors have modified the manuscript to address some of my previous comments. However, there are a couple of remaining issues that I think should be addressed.

1. Regarding the 4L-WS₂ system in Fig. 5b: the authors have clarified that there is unequal relaxation in the innermost vs outermost layers of the system. My understanding is that the relaxation in the outer layers has been ignored for the diffraction simulations. While it makes sense to me that the PLD amplitude in the outer layers is weaker, I am not convinced that the PLD in these layers can simply be ignored. For example, there are additional faint peaks present in the experimental data that are not present in the simulations. The authors should provide simulated diffraction data that includes PLD in the outermost layers to confirm whether additional superlattice peaks arise from the relaxation of those layers.

We now provide simulated diffraction data that includes PLD in the outermost layers to confirm whether additional superlattice peaks arise from the relaxation of those layers. Diffraction of torsional PLDs in 4L-WS₂ with different amplitudes in each layer are now shown in Supplemental Figure 11. To be comprehensive, we make no assumption about the energy landscape and simulate several PLD direction directions. As the reviewer intuited, the outer most superlattice peaks are more consistent with the data when a small PLD is present in the outer layers and Figure 5b has been updated for better accuracy. The conclusions and statements pertaining to Fig.5 remain unchanged—a qualitative demonstration of PLD relaxation behavior across different twisted 2D materials.

Fig. S11 | **Torsional PLDs in 4L-WS₂**: a) Schematic diagram of 4L-WS₂ showing twist configuration of each layer. b) Experimental SAED patterns for 4L-WS₂. c-h) Simulated electron diffraction patterns for 4L-WS₂ with different torsional PLD amplitudes across layers. Accompanying pictographs denotes sign (+/-) and amplitude of torsional PLDs. c, d) PLD amplitudes are weaker (50%) in outer two layers than in inner layers. e, f) PLD amplitudes are equal in all four layers. g) PLDs exists in inner two layers with no PLD in outer layers. h) Simulations without PLDs. c, e) The sign of PLD in upper two and bottom two layers are equal. d, f) PLD sign alternates. In all cases, superlattice peak intensities are qualitatively similar, while some superlattice peaks are different. Experimental SAED patterns (Fig. 5b) matches closely with c) and d).

2.Regarding the WS₂/MoSe₂ system in Fig. 5d: the authors have confirmed longitudinal components are present in heterobilayers. However, I still do not understand the superlattice and Bragg peak orientations. Based on SI Fig. S1b, longitudinal PLDs produce superlattice peaks that extend radially from the primary Bragg peaks, but this does not match what is observed in Fig. 5d. It is also odd that the WS₂ and MoSe₂ appear to have nearly identical lattice constants when these materials have a relatively large lattice mismatch (~4%). Have the experimental diffraction peaks been rotated at all from their original orientations?

If they have been rotated, then this would answer my questions but should be mentioned. If they have not been rotated, then 1) why are the superlattice peaks oriented azimuthally instead of radially with respect to the primary Bragg peaks? Is this because the twist angle is so large? And 2) why do the WS₂ and MoSe₂ Bragg peaks have approximately the same radial distance?

We thank the reviewer for calling attention to a severe typo causing undue confusion. The heterostructure presented in this manuscript is MoSe₂/WSe₂ not WS₂. MoSe₂ and WSe₂ have comparable lattice constants (\lesssim 1%). This explains why the longitudinal component of the PLD is not apparent in figure 5d. This correction has been made and we now also include the full diffraction pattern of the heterostructure as Supplemental Figure 12. We are sincerely grateful for the attention here.

Reviewer Comments, third round -

Reviewer #2 (Remarks to the Author):

The authors have thoughtfully addressed my concerns. I now recommend this manuscript for publication.